# Molecular Epidemiology of Antibiotic Resistance Genes and Virulence Factors in Multidrug-Resistant *Escherichia coli* Isolated from Rodents, Humans, Chicken, and Household Soils in Karatu, Northern Tanzania

**DOI:** 10.3390/ijerph19095388

**Published:** 2022-04-28

**Authors:** Valery Silvery Sonola, Abdul Katakweba, Gerald Misinzo, Mecky Isaac Matee

**Affiliations:** 1Department of Wildlife Management, College of Forestry, Wildlife and Tourism, Sokoine University of Agriculture, P.O. Box 3073, Morogoro 67125, Tanzania; 2Livestock Training Agency (LITA), Buhuri Campus, P.O. Box 1483, Tanga 21206, Tanzania; 3Africa Centre of Excellence for Innovative Rodent Pest Management and Biosensor Technology Development (ACE-IRPM & BTD), Pest Management Institute, Sokoine University of Agriculture, P.O. Box 3110, Morogoro 67125, Tanzania; katakweba@sua.ac.tz; 4Institute of Pest Management, Sokoine University of Agriculture, P.O. Box 3110, Morogoro 67125, Tanzania; 5Department of Veterinary Microbiology, Parasitology and Biotechnology, College of Veterinary Medicine and Biomedical Sciences, Sokoine University of Agriculture, P.O. Box 3297, Morogoro 67125, Tanzania; gerald.misinzo@sacids.org; 6SACIDS Foundation for One Health, Sokoine University of Agriculture, P.O. Box 3297, Morogoro 67125, Tanzania; mateemecky@yahoo.com; 7Department of Microbiology and Immunology, Muhimbili University of Health and Allied Sciences, P.O. Box 65001, Dar es Salaam 11103, Tanzania

**Keywords:** multidrug-resistant, rodents, chicken, humans, soil, *E. coli*, PCR, genes

## Abstract

The interaction of rodents with humans and chicken in the household environment can facilitate transmission of multidrug-resistant (MDR) *Escherichia coli* (*E. coli*), causing infections that are difficult to treat. We investigated the presence of genes encoded for carbapenem, extended spectrum beta-lactamases (ESBL), tetracycline and quinolones resistance, and virulence among 50 MDR *E. coli* isolated from human (*n* = 14), chicken (*n* = 12), rodent (*n* = 10), and soil (*n* = 14) samples using multiplex polymerase chain reaction (PCR). Overall, the antimicrobial resistance genes (ARGs) detected were: *blaTEM* 23/50 (46%), *blaCTX-M* 13/50 (26%), *tetA* 23/50 (46%), *tetB* 7/50 (14%), *qnrA* 12/50 (24%), *qnrB* 4/50 (8%), *blaOXA-48* 6/50 (12%), and *blaKPC* 3/50 (6%), while *blaIMP*, *blaVIM*, and *blaNDM-1* were not found. The virulence genes (VGs) found were: *ompA* 36/50 (72%), *traT* 13/50 (26%), *east* 9/50 (18%), *bfp* 5/50 (10%), *eae* 1/50 (2%), and *stx-1* 2/50 (4%), while *hlyA* and *cnf* genes were not detected. Resistance (*blaTEM*, *blaCTX-M*, *blaSHV*, *tetA*, *tetB*, and *qnrA*) and virulence (*traT*) genes were found in all sample sources while *stx-1* and *eae* were only found in chicken and rodent isolates, respectively. Tetracycline resistance phenotypes correlated with genotypes *tetA* (r = 0.94), *tetB* (r = 0.90), *blaKPC* (r = 0.90; *blaOXA-48* (r = 0.89), and *qnrA* (r = 0.96). ESBL resistance was correlated with genotypes *blaKPC* (r = 0.93), *blaOXA*-48 (r = 0.90), and *qnrA* (r = 0.96) resistance. Positive correlations were observed between resistance and virulence genes: *qnrB* and *bfp* (r = 0.63) also *blaTEM*, and *traT* (r = 0.51). Principal component analysis (PCA) indicated that *tetA*, *tetB*, *blaTEM*, *blaCTX-M*, *qnrA*, and *qnrB* genes contributed to tetracycline, cefotaxime, and quinolone resistance, respectively. While *traT stx-1*, *bfp*, *ompA*, *east*, and *eae* genes contributed to virulence of MDR *E. coli* isolates. The PCA ellipses show that isolates from rodents had more ARGs and virulence genes compared to those isolated from chicken, soil, and humans.

## 1. Introduction

*Escherichia coli* (*E. coli*) is a versatile bacterial pathogen that has the ability to cause various infections, most of which are difficult to treat [1,2]. In fact, this bacterium is listed by the World Health Organization (WHO) as one of the critical antimicrobial-resistant bacteria that can cause severe and often deadly infections such as bloodstream infections and pneumonia [3]. The pathogenicity of *E. coli* strains is enhanced by a variety of virulence and resistance genes [4,5]. *E. coli* strains producing extended-spectrum β-lactamases (ESBLs) and carbapenemases are potentially recognized pathogens that can resist most β-lactam antibiotics [6,7]. ESBLs are plasmid-mediated enzymes that hydrolyse β-lactam containing antimicrobial agents including penicillins, cephalosporins, and aztreonam. ESBLs are grouped into three main types: TEM, SHV, and CTX-M [8,9]. Carbapenemases are a major group of β-lactamases capable of hydrolysing penicillins, cephalosporins, monobactams, and carbapenems. They include β-lactamases of classes B (IMP and VIM), D (OXA-23 to -27), and A (IMI, KPC, NMC, and SME) [10,11]. Tetracycline resistance genes (*tetA* and *tetB*) coded for efflux pumps have been frequently detected in human and animal *E. coli* isolates [12]. The genes *qnrA* and *qnrB* are known to confer quinolone resistance in *E. coli* strains and spread horizontally through plasmids [13].

Important virulence factors of *E. coli* are encoded by several genes including: locus enterocyte effacement (LEE), intimin, bundle forming (*bae*, *bfpA*)) [14,15], Shiga toxins, adhesins (*stx1*, *stx2*, *eaeA*, *ehxA, and bfpA*) [15,16], heat-labile, heat stable, and colonization factors (*elt*, *est)* [14,16]. *E. coli* is a typical One Health pathogen, with the potential of resistomes spreading between humans, animals, and the environment, where such interactions exist [17]. In Tanzania, studies conducted in the Karatu ecosystem have revealed intense interactions between humans, rodents, and chicken, leading to frequent occurrence and recurrence of zoonotic infections [18,19,20]. Previous studies have suggested that the role of rodents in the transmission of multidrug-resistant (MDR) bacterial infections to humans and environmental contamination [21,22,23,24]. In a recent phenotypic study conducted in Karatu, we isolated *E. coli* strains from chickens, humans, rodents, and soils which showed high levels of resistance to cefotaxime (79.7%), imipenem (79.8%), and tetracycline (73.7%); 512 out of 650 (78.8%) were MDR [25].

We hypothesize that the intense interactions between chickens, humans, rodents, and soils may lead to the transfer of ARGs and VRGs among them. However, molecular characterization of ARGs and VGs was not conducted in the phenotypic study [25]. Knowledge of ARGs and VGs is important in understanding the pathogenicity and virulence of *E. coli* [26]. This study was conducted in Karatu, Northern Tanzania to provide insights of molecular epidemiology of ARGs and VGs occurring in *E. coli* isolated among chickens, humans, rodents, and soils in households. To our knowledge, this is the first study in Tanzania that has investigated the genotypic diversity of *E. coli* isolated among chicken, humans, rodents, and soils in households. Multiplex PCR [27] was used for detection of genes encoding for tetracycline resistance (*tetA*, *tetB*), ESBL (*blaCTX-M*, *blaSHV*, and *blaTEM*), metallo beta-lactamases (*blaVIM*, *blaIMP*, and *blaNDM*), and virulence genes *bfp*, *east*, *hlyA*, *traT*, *eaeA*, *ompA*, *cnf*, and *stx-1.* The working assumption is that MDR *E. coli* strains circulating in Karatu carry a variety of virulence genes capable of causing life-threatening infections that are difficult to treat.

## 2. Materials and Methods

### 2.1. Study Area

This study was conducted between June 2020 and March 2021 in the Karatu district in the northern zone of Tanzania, located between latitudes 3°10′ and 4°00′ S, and longitude 34°47′ and 35°56′ E. The district has a population of 230,166 people comprised of 117,769 men and 112,397 women with an average of five people per household [28]. Karatu has an altitude range from 1000 to 1900 m above sea level with two wet seasons annually (short rains between October and December and long rains from March to June).

### 2.2. Bacterial Isolates

A total of 50 MDR *E. coli* isolates from chicken cloaca swabs (12), human nasal swabs (14), rodents’ deep pharyngeal swabs (10), and household soil (14) samples, particularly those with higher phenotypic resistance to tetracycline, imipenem, and cefotaxime, were selected for genomic DNA extraction and further genomic analyses. All selected isolates were preserved in nutrient broth (TSB) with 50% glycerol (*v*/*v*) at −80 °C until DNA extraction. Isolates that were resistant to at least three different classes of antibiotics were considered as multidrug-resistant (MDR) [29].

### 2.3. DNA Extraction

The genomic DNA of all phenotypically MDR *E. coli* strains were extracted by using Zymo Research Fungal and Bacterial Genomic DNA MiniPrep^TM^ kit (Zymo Research, Irvine, CA, USA), following the manufacturer’s instructions. The purity, quality, and quantity of DNA were determined using a nanodrop spectrophotometer (NanoDrop, Thermo Scientific, Ramsey, NJ, USA) and agarose gel electrophoresis. The extracted DNA samples were stored at −80 °C until when PCR analyses were performed.

### 2.4. Detection of Antimicrobial Resistance and Virulence Genes

Multiplex PCR [27] was used to detect the tetracycline (*tetA* and *tetB*), ESBL (*blaCTX-M*, *blaSHV*, and *blaTEM*), and Metallo beta-lactamases (*blaVIM*, *blaIMP*, and *blaNDM*) resistance and virulence (*bfp*, *east*, *hlyA*, *traT*, *eae*, *ompA*, *cnf*, and *stx1*) genes. Briefly, lyophilized primers (Macrogen, Amsterdam, The Netherlands) for target genes (in Appendix A) were reconstituted using nuclease-free water to obtain 100 μM stock and 10 μM working solutions before storage at −20 °C. PCR was carried out in a total volume of 25 μL containing 12.5 μL of 1 X *Taq* PCR Master Mix (Bio Basic, Canada), 1 μL of the forward primer and 1 μL of the reverse primer, 3 μL of DNA template, and 7.5 μL nuclease-free water. Multiplex PCRs were conducted using amplification conditions indicated in Table 1. PCR products were separated by electrophoresis on 1.5% (*w*/*v*) agarose gel pre-stained with Gel Red (Merck, Darmstadt, Germany) at 120 Volts for 1 h, and visualized under UV light using a BioDoc-it^TM^ imaging system (Ultra-Violet Products, Cambridge, UK). PCR product size was determined by conducting electrophoresis along with a GeneRuler 100 bp Plus DNA Ladder (Bioneer, Daedeok-gu, Republic of Korea). DNA from *E. coli* American Type Culture Collection (ATCC) 29522 strain was used for quality assurance.

### 2.5. Statistical Analysis

The data obtained were entered into an Excel spreadsheet (Microsoft^®^ Office Excel 2010) and analysed. The differences in occurrence of the genes (%) between categories were compared by chi-square test using R-software, version 4.0.2 (R Foundation for Statistical computing, Vienna, Austria) [30]. Principal component analysis (PCA) was used to investigate the distribution and relationships of antimicrobial resistance and virulence genes of MDR *E. coli* isolates with respect to their different sample sources. Any *p*-value less than 0.05 was considered statistically significant.

## 3. Results

### 3.1. Carbapenems, ESBL, Tetracycline, and Quinolones Resistance Genes in MDR E. coli Isolates from Different Sample Sources

Overall, the resistance genes *blaTEM* (46%), *blaCTX-M* (26%), *tetA* (46%), *tetB* (14%), *qnrA* (24%), *qnrB* (8%), *blaOXA-48* (12%), and *blaKPC* (6%) were detected (Figure 1) and distributed in isolates from human 8/14 (57.1%), chicken 9/12 (75.0%), rodent 8/10 (80.0%), and soil 7/14 (50.0%) samples, as shown in Table 2. For human isolates the most common ARGs were *tetA* 8/14 (57.1%) and *blaSHV* 5/14 (35.7%), while for chicken the most common ones were *tetA* 5/12 (41.7%) and *qnrA* 6/12 (50%), for rodents they were *blaTEM* 6/12 (50%), and *tetA* 6/12 (50%), and for soil they were *blaTEM* 7/14 (50%) and *tetA* 4/14 (28.6%).

### 3.2. Detection of Virulence Genes in MDR E. coli Isolates from Different Sample Sources

Overall, the virulence genes were: *ompA* (72%), *traT* (26%), *east* (18%), *bfp* (10%), *eae* (2%), and *stx-1* (4%), while *hlyA* and *cnf* were not detected (Table 1). For humans the most common VRs were *traT* 4/14 (28.6%) and *ompA* 10/14 (71.4%), while for chicken they were *traT* 4/12 (33.3%) and *ompA* 12/12 (100%), for rodents they were *traT* 4/10 (40%) and *ompA* 7/10 (70%), and for soil isolates they were predominated by *ompA* 7/14 (50%) and *east* 2/14 (14.3) (Figure 2).

### 3.3. Comparison between Phenotypic and Genotypic Antibiotic Resistance

We found positive correlations between tetracycline resistance and *tetA* (0.94), *tetB* (=0.90), carbapenem resistance and *blaKPC* (0.90) and *blaOXA-48* (0.89), and quinolone resistance and *qnrA* (0.96). We also found correlation between tetracycline resistance and genotypes for carbapenem (*blaKPC* = 0.90, *blaOXA-48* = 0.91), cefotaxime and *qnrA* (0.96), and quinolone resistance and *qnrA* (0.94). Cefotaxime resistance was correlated with genotypes for carbapenem (*blaKPC* = 0.93, *blaOXA-48* = 0.90) and quinolone (*qnrA* = 0.96) resistance (Table 3). However, we found weak and negative correlation between phenotypes and genotypes for ESBL resistance (*CTX-M* = 0.60, *blaTEM* = −0.63 and *blaSHV* = 0.33) (Table 3).

As shown in Table 4, we found correlations between *qnrB* and *bfp* genes (r = 0.63) and with *blaTEM* and *traT* genes (r = 0.51) and the remaining displayed weak and negative correlations.

### 3.4. Co-Occurrence between Resistance and Virulence Genes

We observed that 38 out of 50 (76%) MDR *E. coli* isolates had at least one virulence gene. A co-existence of up to six resistance genes and at least one virulence gene was noted. In some cases, four resistance genes co-existed with four virulence genes (Figure 3). The combination consisting of *blaTEM*, *blaCTX-M*, *tetA*, *ompA*, and *traT* genes was common with 55% co-occurrence (Figure 4 and Figure 5).

### 3.5. Principal Component Analysis Results

According to Figure 6 below, the arrows (vectors) for *tetA*, *qnrA*, and *tetB* genes aligned closer to each other in principal component 1 (PC1) indicating greater and positive correlations among them. The lengths of arrows show that *tetB* gene contributed more to the resistance of isolates followed by *qnrB* and *tetA* genes. The vectors for *blaTEM*, *blaCTX-M*, and *qnrB* genes are close to each other and to PC2 showing their influence on resistance. These genes had greater and positive correlations between them, but all were negatively correlated to the *blaSHV* gene. The lengths of the vectors indicate that the *blaTEM* gene had a higher influence on resistance of isolates (PC2), followed by the *blaCTX-M* while *qnrB* had the lowest. According to PCA plane, rodent and chicken ellipses are extended in the upper quadrants indicating higher proportions of ARGs, followed by those from human and soil.

The smaller angle between *traT* gene vector and PC1 indicates a greater and positive correlation between them (Figure 7). The same behaviour was displayed by *east* and *eae* genes which show a greater and positive correlation between them. Along PC2, *stx-1*, *bfp* and *ompA* genes had greater and positive correlations with PC2 indicating higher influence on virulence of isolates. The different sizes of loadings indicated higher and positive correlations between them. Different sizes of ellipses indicate variation in the prevalence of virulence genes across different sources of isolates. Rodent isolates had more virulence genes followed by chicken and soil isolates, while those from humans had the lowest gene prevalence.

## 4. Discussion

The study found 32/50 (64%) of MDR *E. coli* isolates carrying at least one AMR gene, with 10/50 (20%) having more than three. At the same time, 38 out of 50 (76%) MDR *E. coli* isolates had at least one virulence gene and 8/50 (16%) had more than three. PCA results showed that most of the resistance and virulence genes were found in isolates from rodents and chicken samples compared with human and soil isolates (Figure 6 and Figure 7). The most detected AMR genes included: *tetA* (46%), *blaTEM* (46%), *blaCTX-M* (26%), *qnrA* (24%), *blaSHV* (22%), *tetB* (8%), and *blaOXA-48* (12%). This finding is in agreement with the results of our previous study in Karatu that reported higher resistance of *E. coli* to tetracycline (73.7%), imipenem (79.8%), and cefotaxime (79.7%) where 512 out of 650 (78.8%) isolates were multidrug-resistant [25]. Interestingly, the highest prevalence of AMR genes was observed in isolates from rodent (80.0%) followed by those from chicken (75.0%), human (57.1%), and lastly soil (50.0%) samples. Our findings imply that rodents that invade households have a potential to spread MDR *E. coli* infections with ARGs to other hosts, as observed by others [31,32,33]. The increased prevalence of resistance genes in isolates from chicken can be associated with frequent use and misuse of antibiotics in the prevention and treatment of poultry diseases, which is a common practice in the area as reported by previous studies [26,34,35]. The high prevalence of ESBL genes; *blaSHV* (20%), *blaCTX-M* (40%), *blaTEM* (60%), tetracycline; *tetA* (60%), and quinolone; *qnrB* (20%) resistance genes indicate the widespread of MDR *E. coli* infections in the Karatu district. This keeps with findings of a study conducted in nearby Arusha that found *blaTEM*, *blaCTX-M*, *tetA*, *tetB*, and *qnrs* [34,35,36]. This pattern can be explained by the frequent use and misuse of these antibiotics in veterinary and human medicines in the area [35], rendering these groups of antibiotics to be less effective. We found strong and positive correlations between tetracycline resistance and *tetA* (r = 0.94) and *tetB* (r = 0.90), carbapenem resistance and *blaKPC* (r = 0.94), as well as *blaOXA-48* (r = 0.89) and quinolone resistance with *qnrA* (r = 0.96), highlighting the dominant role of genes in causing resistance [37,38,39]. Similarly, we found strong and positive correlation between tetracycline resistance phenotypes and genotypes for carbapenem (*blaKPC* = 0.90, *blaOXA-48* = 0.91), quinolone (*qnrA* = 0.94), as well as ESBL and carbapenem (*blaKPC* = 0.93, *blaOXA-48* = 0.90) and quinolone (*qnrA* = 0.96) resistance genotypes. Such associations have been reported in previous studies [40,41] and can be explained by the fact that most of these genes are carried on similar transferrable plasmids [42,43].

Overall, the detected virulence genes were: *bfp* 5/50 (10%), *east* 9/50 (18%), *traT* 13/50 (26%), *eae* 1/50 (2%), *ompA* 36/50 (72%), and *stx-1* 2/50 (4%). For isolates obtained from human samples, the most common virulence genes were: *traT* (28.6%) and *ompA* (71.4%), for chickens *ompA* (100%), *traT* (33.3%), *east* (33.3%), and *stx1* (8.3%) for rodents *ompA* (70%), *eae* (10%), *traT* (40%), *east* (30%), *bfp* (30%), and *stx1* (10%). Isolates from soil samples contained *bfp* (14.3%), *east* (14.3%), *traT* (7.1%), and *ompA* (50%). The bundle forming pilus (*bfp*) gene codes for adherence of *E. coli* strains to intestinal epithelial cells of the host [44], while *eae* gene promotes secretion of intimin protein for bacterial adherence to enterocytes [45]. The gene stx-1 encodes production of the Shiga toxin (stx) protein in some *E. coli* strains responsible for haemolytic uremic syndrome (HUS) and bloody diarrhoea in humans [45,46]. The gene *east* codes for production of heat-stable enterotoxin 1 in Enteroaggregative *E. coli* (EAST1) which induces diarrhoea in humans and livestock [47]. The gene *ompA* codes for outer membrane protein A, which enables intracellular survival of *E. coli* strains and protects them against host defence mechanism [48]. Meanwhile the *traT* gene codes for outer membrane protein, an important factor during urethral tract infections in humans [49]. The presence of wide-ranging virulence factors indicates that the MDR *E. coli* isolates circulating in Karatu have the ability to cause life-threatening infections that can be difficult to treat, given the fact that they occur in antibiotic-resistant isolates. We noted some significant differences with other studies. In this study, the prevalence of *ompA* in rodent isolates (70%) was lower than 93.5% reported in China by Guan et al. [50]. The 50% occurrence of *ompA* in *E. coli* from soil samples was greater than 42% documented in Indiana, USA [51]. However, we did not detect *stx1*, *eae*, and *hlyA* genes contrary to Cooley et al. [52] who reported *stx1* (100%), *eae* (100%), and *hlyA* (40%) in soil, livestock, wild birds, and water samples, respectively. Interestingly, we found a higher prevalence of virulence genes (60%) among *E. coli* isolates from rodent samples compared to previous studies in Berlin (0%) [21], in Hanoi (1.7%) [53], and in Vancouver (3.8%) [54]. These geographical related differences can be attributed to variations in levels of antibiotics use as well as environmental factors [55]. In this study, we found co-occurrence of resistance and virulence genes in 38/50 (76%) of the isolates. The most common combinations were: *blaSHV*, *tetA*, and *ompA* in humans; *blaTEM*, *tetA*, *tetB*, *qnrA*, and *ompA* in chicken; *blaTEM*, *blaCTX-M*, *tetA*, and *ompA* in rodents; and *blaTEM*, *tetA*, and *ompA* in soil isolates. Importantly, we found varying correlation between ARGs and VGs among the isolates. We found positive correlation between *blaTEM* and *traT* genes (r = 0.51) and *qnrB* and *bfp* genes (r = 0.63), while negative correlations were revealed between *blaOXA-48* and *ompA* (r = −0.05), *blaSHV* and *traT* (r = −0.44), and *tetA* and *east* (r = −0.10). This finding is keeping with those of other studies, showing that acquisition of resistance to certain antimicrobial agents may be associated with an increase or decrease in the virulence levels of a microorganism. This result seems to indicate that acquisition of resistance to certain antibiotics may be associated with an increase or decrease in the virulence levels depending on location and mechanism of transfer of specific genes [27,56,57].

## 5. Conclusions

Our study revealed that MDR *E. coli* isolates from humans, chicken, rodents, and household soils harbour different ARGs (*blaTEM*, *blaCTX-M*, *blaSHV*, *tetA*, *tetB*, *qnrA*, and *qnrB*) and VGs (*bfp*, *east*, *traT*, *ompA* and *stx-1*). The PCA results show that *traT*, *stx-1*, *bfp*, *ompA*, *east*, and *eae* genes influenced the virulence of MDR *E. coli* isolates. Resistance (*blaTEM*, *blaCTX-M*, *blaSHV*, *tetA*, *tetB*, and *qnrA*) and virulence (*traT*) genes were detected in isolates from all sample sources, while *stx-1* and *eae* genes were specific to chicken and rodent isolates only. Interestingly, rodents had the highest percentage of both ARGs and VGs, indicating their potential in carriage and transmission of infections to other hosts in the environment. This situation urgently calls for One Health-based interventions including improving hygiene and control of rodents in households.

## Figures and Tables

**Figure 1 ijerph-19-05388-f001:**
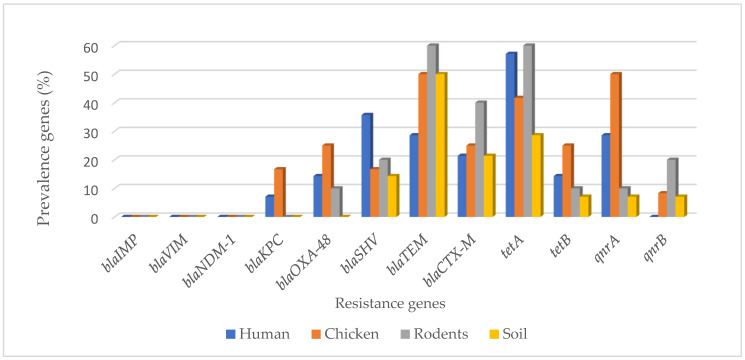
Occurrence of resistance genes in MDR *E. coli* isolates from different sample types.

**Figure 2 ijerph-19-05388-f002:**
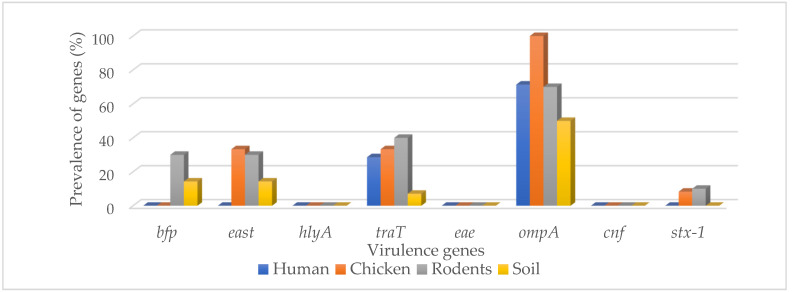
Prevalence of virulence genes in different types of the sample source.

**Figure 3 ijerph-19-05388-f003:**
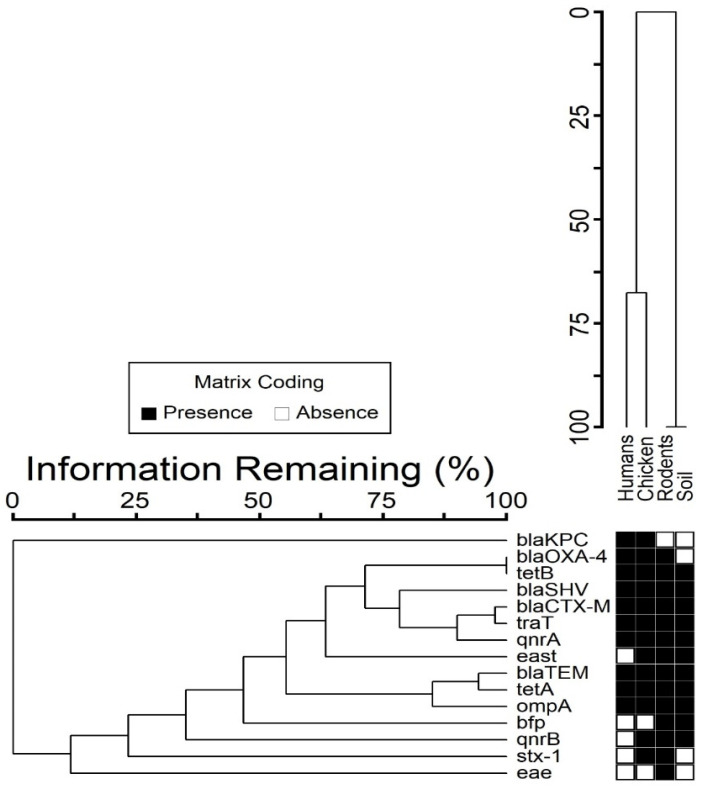
Co-occurrence of resistance and virulence genes in isolates from different sample sources.

**Figure 4 ijerph-19-05388-f004:**
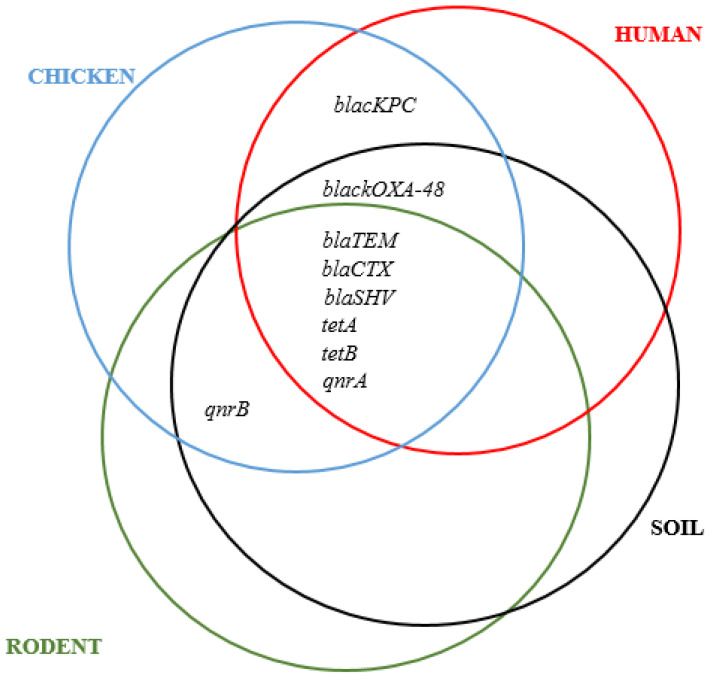
Distribution of resistance genes in various sample sources.

**Figure 5 ijerph-19-05388-f005:**
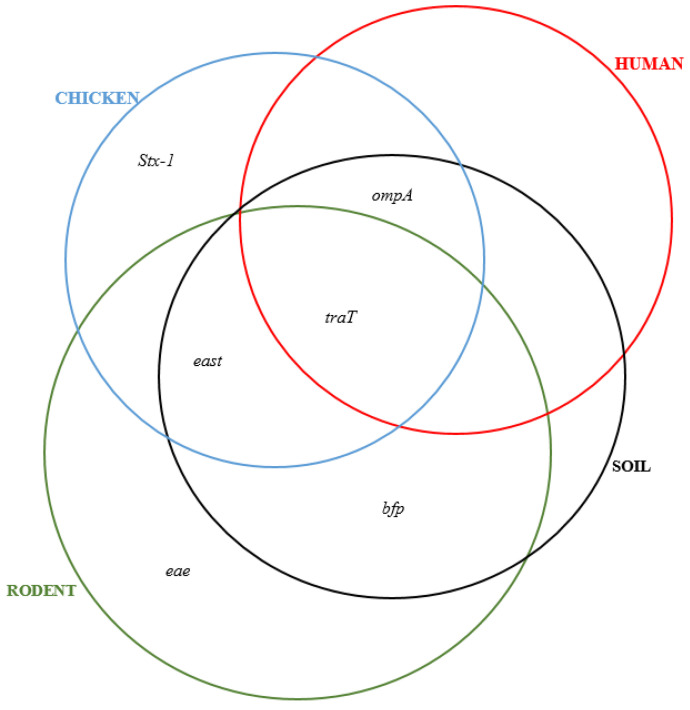
Distribution of virulence genes in various sample sources.

**Figure 6 ijerph-19-05388-f006:**
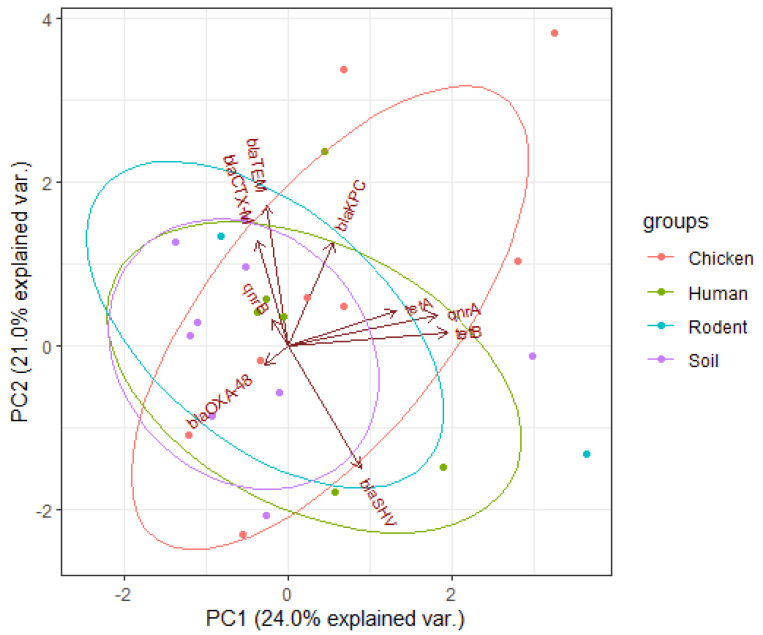
Principal component analysis of resistance genes of *E. coli* isolates. The dots represent isolates from different sources of samples, arrows indicate the original variables (resistance genes of the isolates), and ellipses indicate a region that contains 95% of all samples of a particular source.

**Figure 7 ijerph-19-05388-f007:**
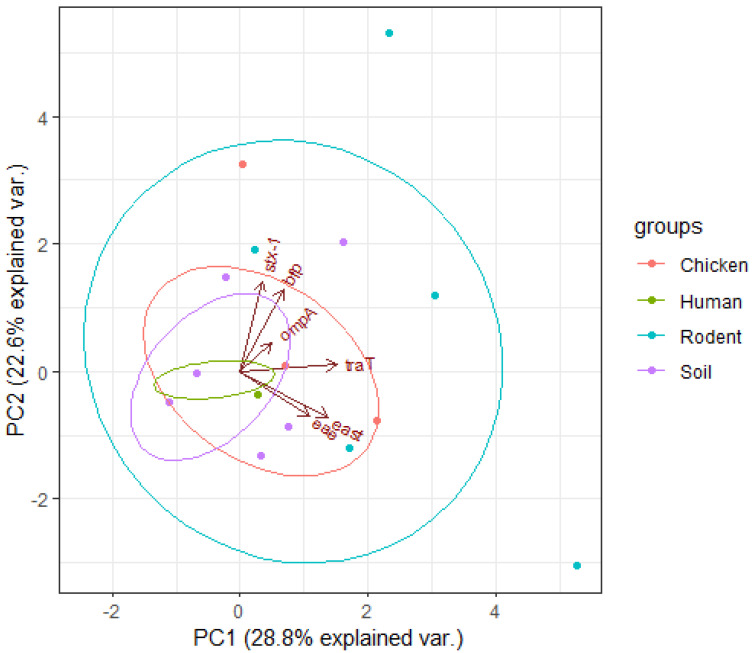
Principal component analysis for virulence genes of *E. coli* isolates. The dots represent isolates from different sources of samples, arrows indicate the original variables (virulence genes of the isolates), and ellipses indicate a region that contains 95% of all samples of a particular source.

**Table 1 ijerph-19-05388-t001:** Detection of virulence genes of MDR *E. coli* isolates from different sample sources.

Genes	Different Sample Sources *n* (%)				
	Humans (*n* = 14)	Chickens (*n* = 12)	Rodents (*n* = 10)	Soil (*n* = 14)	Total (*n* = 50)
*Bfp*	0 (0.0)	0 (0.0)	3 (30.0)	2 (14.3)	5 (10.0)
*East*	0 (0.0)	4 (33.3)	3 (30.0)	2 (14.3)	9 (18.0)
*hlyA*	0 (0.0)	0 (0.0)	0 (0.0)	0 (0.0)	0 (0.0)
*traT*	4 (28.6)	4 (33.3)	4 (40.0)	1 (7.1)	13 (26.0)
*eae*	0 (0.0)	0 (0.0)	1 (10.0)	0 (0.0)	1 (2.0)
*ompA*	10 (71.4)	12 (100.0)	7 (70.0)	7 (50.0)	36 (72.0)
*cnf*	0 (0.0)	0 (0.0)	0 (0.0)	0 (0.0)	0 (0.0)
*stx-1*	0 (0.0)	1 (8.3)	1 (10.0)	0 (0.0)	2 (4.0)
Total	2 (14.3)	4 (33.3)	6 (60.0)	4 (33.3)	16 (32.0)
χ^2^-square	52.29	46.43	2.00	26.67	
*p*-value	0.001	0.001	0.0188	0.0004	

**Table 2 ijerph-19-05388-t002:** Prevalence of antimicrobial resistance genes in MDR *E. coli* isolates from different sample types.

Genes	Different Types of Sample Sources *n* (%)				
	Human (*n* = 14)	Chicken (*n* = 12)	Rodents (*n* = 10)	Soil (*n* = 14)	Total Isolates (*n* = 50)
*blaIMP*	0 (0.0)	0 (0.0)	0 (0.0)	0 (0.0)	0 (0.0)
*blaVIM*	0 (0.0)	0 (0.0)	0 (0.0)	0 (0.0)	0 (0.0)
*blaNDM-1*	0 (0.0)	0 (0.0)	0 (0.0)	0 (0.0)	0 (0.0)
*blaKPC*	1 (7.1)	2 (16.7)	0 (0.0)	0 (0.0)	3 (6.0)
*blaOXA-48*	2 (14.3)	3 (25.0)	1 (10.0)	0 (0.0)	6 (12.0)
*blaSHV*	5 (35.7)	2 (16.7)	2 (20.0)	2 (14.3)	11 (22.0)
*blaTEM*	4 (28.6)	6 (50.0)	6 (60.0)	7 (50.0)	23 (46.0)
*blaCTX-M*	3 (21.4)	3 (25.0)	4 (40.0)	3 (21.4)	13 (26.0)
*tetA*	8 (57.1)	5 (41.7)	6 (60.0)	4 (28.6)	23 (46.0)
*tetB*	2 (14.3)	3 (25.0)	1 (10.0)	1 (7.1)	7 (14.0)
*qnrA*	4 (28.6)	6 (50.0)	1 (10.0)	1 (7.1)	12 (24.0)
*qnrB*	0 (0.0)	1 (8.3)	2 (20.0)	1 (7.1)	4 (8.0)
Total	8 (57.1)	9 (75.0)	8 (80.0)	7 (50.0%)	32 (64.0)
χ^2^-square	52.29	46.43	2.00	26.67	
*p*-value	0.001	0.001	0.0188	0.0004	

**Table 3 ijerph-19-05388-t003:** Correlation between phenotypes and genotypes of MDR *E. coli* isolates.

Genotypes of Isolates	Phenotypic Resistance of Isolates			
	Correlation Coefficients (r)			
	Tetracycline	Imipenem	Ciprofloxacin	Cefotaxime
*tetA*	0.53	0.51	0.62	0.43
*tetB*	0.90	0.90	0.86	0.93
*blaKPC*	0.90	0.90	0.86	0.93
*blaOXA-48*	0.91	0.89	0.90	0.90
*qnrA*	0.94	0.94	0.90	0.96
*qnrB*	−0.69	−0.71	−0.67	−0.69
*blaCTX-M*	−0.54	−0.58	−0.45	−0.61
*blaTEM*	−0.71	−0.69	−0.78	−0.63
*blaSHV*	0.43	0.41	0.51	0.33

**Table 4 ijerph-19-05388-t004:** Correlation between resistance and virulence genes of MDR *E. coli* isolates.

ABR Genes	Virulence Genes					
	Correlation Coefficients (r)					
	*bfp*	*east*	*traT*	*eae*	*ompA*	*stx-1*
*blaKPC*	−0.11	0.30	0.41	−0.05	0.11	−0.07
*blaOXA-48*	−0.16	−0.06	0.15	0.38	−0.05	0.22
*blaSHV*	−0.24	−0.34	−0.44	−0.10	0.24	−0.15
*blaTEM*	0.39	0.44	0.51	0.17	−0.39	0.01
*blaCTX-M*	0.05	0.39	0.31	0.23	−0.22	0.08
*tetA*	0.36	−0.10	0.11	0.15	0.26	0.22
*tetB*	−0.18	0.06	−0.05	−0.08	0.18	−0.11
*qnrA*	−0.26	0.03	0.00	−0.11	0.09	0.10
*qnrB*	0.63	−0.19	0.12	−0.05	0.13	0.30

## Data Availability

The data presented in this study are available on request from the corresponding author. The data are not publicly available due to privacy restrictions.

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
