# Peer review of "Molecular Epidemiology of Antibiotic Resistance Genes and Virulence Factors in Multidrug-Resistant Escherichia coli Isolated from Rodents, Humans, Chicken, and Household Soils in Karatu, Northern Tanzania"

_ijerph, 2022, doi:10.3390/ijerph19095388_

Round 1

Reviewer 1 Report

The manuscript described the Molecular Epidemiology of Carbapenems, ESBL, Tetracycline and Quinolones Resistance Genes and Virulence Factors in 
Multidrug-Resistant Escherichia coli Isolated From Humans, Rodents, Chickens and Household Soils in Karatu, Northern Tanzania. The authors found that isolates from rodents had more ARGs and virulence genes compared to those isolated from chicken, soil, and humans. Antimicrobial resistance is a global and cross-sectoral issue that needs attention from all stakeholders for effective mitigation. The introduction section contains several biased and controversial comments which should be modified and divided into 2-3 paragraphs. The discussion section should be improved enough with proper justifications of the findings, not a mere comparison with earlier studies and repetition of results.

Comment 1. The introduction section should be divided into 2-3 paragraphs and rewrite the gene's names correctly.

Comment 2. M&M section: Bacterial isolates: "Isolates that were resistant to at least three different classes of antibiotics were considered as multidrug-resistant (MDR)"..... Reference is missing

Comment 3. The authors need to clarify the samples collection. What kind of samples were isolated; fecal, blood, tissue, etc.? Please add the details.

Comment 4. Detection of antimicrobial resistance and virulence genes: Why did the authors only choose only specific resistance genes against four drugs, what were the criteria of selection only for these ARGs and VGs? Also, there are many virulence genes associated with E. coli then why only a few were selected as all animals have different kinds of virulence genes?

Comment 5. Antibiotic susceptibility testing results of these 50 multidrug-resistant E. coli are missing.

Comment 6. The discussion section should be improved enough with proper justifications of the findings, not a mere comparison with earlier studies and repetition of results. You can also take help from the following related studies: 

    • DOI: 10.1292/jvms.19-0697;
    •     DOI: 10.1016/j.micpath.2020.104722;
    • DOI: 10.1111/jam.15469;    DOI: 10.1038/s41598-019-46831-3
  •  

Comment 7. Tables 1 & 2 are not appropriate here. Better to change into supplementary.

Reviewer 2 Report

The manuscript entitled 'Molecular epidemiology of Carbapenems, ESBL, tetracycline and quinolones re-sistance genes and virulence factors in Multidrug-resistant Escherichia coli isolated from Humans, Rodents, Chickens and Household Soils in Karatu, Northern Tanzania ' presented a good study regarding the Detection of  Multidrug-resistant Escherichia coli from ONE-Health perspective. The study is well-designed. However, there is fair bit of clarifications to do so the data and interpretations are tight.

Detail comments:
1. The title is too long. no line number found in the first three pages.
2. Introduction: why other tet resistance genes were not selected, which is critical and is missing
3. Material and Methods: strain information such as the strain name, isolation source is needed to be presented in a table. 
4. line 19: should be ATCC29522 
5. line 23: the reference for R-software and version number is needed. The method for correlation analysis is needed, which is critical and is missing.
6. line 75: there are two Figure 4. And the figure legend for first Figure 4 is too simple.
7. line 77: for the second Figure 4, where is other resistance genes, such as qnrB?
8. line 153: more discussion about the prevelance of Multidrug-resistant Escherichia coli in Chickens is needed.

Reviewer 3 Report

The manuscript title the “Molecular Epidemiology of Carbapenems, ESBL, Tetracycline and Quinolones Resistance Genes and Virulence Factors in Multidrug-Resistant Escherichia coli Isolated From Humans, Rodents, Chickens and Household Soils in Karatu, Northern Tanzania” is a valuable paper. The article should be prepared according to “Instructions for Authors". In all article is a problem with page number, line number and a number of sections, double space and scientific language. References should be prepared according to “Instructions” and “Abstract” in my opinion is too long. A lot of valuable work was done but this article needs extensive language correction.

Now I have only minor comments:

Minor comments:

Abstract;

 after “Escherichia coli” put abbr. (E. coli)

Introduction:

after “Escherichia coli” put abbr. (E. coli)

“bacterium” for me something strange, the Authors should check if it is well

“ESBLs” a double space

“study [26]” a double space

“E. coli [27].” A double space

“2. Materials and Methods”

“Bacterial isolates” isolates or strains ?????

Results;

“Comparison between phenotypic and genotypic antibiotic resistance 53

We found positive correlations between phenotypic tetracycline resistance and tetA 54

 (0.94), tetB (=0.90), carbapenem resistance and blaKPC (0.90) and blaOXA-48 (0.89) and 55

quinolone resistance and qnrA (0.96). We also found correlation between tetracycline re- 56

 sistance and genotypes for carbapenem (blaKPC=0.90, blaOXA-48=0.91), ESBL and quino- 57

 lone resistance (qnrA=0.94). ESBL resistance was correlated with genotypes for car- 58

bapenem (blaKPC=0.93, blaOXA-48=0.90) and quinolone (qnrA=0.96) resistance (Table 5). 59

However, we found weak and negative correlation between phenotypes and genotypes 60

for ESBL resistance (CTX-M= 0.60, blaTEM= -0.63 and blaSHV=0.33) (Table 5). 61

the language should be changed

Round 2

Reviewer 1 Report

The authors adequately answered my questions. However, they just didn't revise my previous question about the details of the samples. What kind of samples were isolated; fecal, blood, tissue, etc.? 

Response 3: Types of samples have been clarified on lines 104 and 105.

I checked the old manuscript with the revised manuscript but the authors did not revise my previous comment no.3.
